# Tactile sensory channels over-ruled by frequency decoding system that utilizes spike pattern regardless of receptor type

Ingvars Birznieks[1,2,3]*, Sarah McIntyre[2,3,4], Hanna Maria Nilsson[2,4], Saad S Nagi[4,5], Vaughan G Macefield[2,5,6], David A Mahns[5], Richard M Vickery[1,2]

[1]School of Medical Sciences, Faculty of Medicine, UNSW Sydney, Sydney, Australia; [2]Neuroscience Research Australia, Sydney, Australia; [3]Biomedical Engineering and Neuroscience, MARCS Institute, Western Sydney University, Sydney, Australia; [4]Linköping University, Linköping, Sweden; [5]School of Medicine, Western Sydney University, Sydney, Australia; [6]The Baker Heart and Diabetes Institute, Melbourne, Australia

**Abstract** The established view is that vibrotactile stimuli evoke two qualitatively distinctive cutaneous sensations, flutter (frequencies < 60 Hz) and vibratory hum (frequencies > 60 Hz), subserved by two distinct receptor types (Meissner's and Pacinian corpuscle, respectively), which may engage different neural processing pathways or channels and fulfil quite different biological roles. In psychological and physiological literature, those two systems have been labelled as Pacinian and non-Pacinian channels. However, we present evidence that low-frequency spike trains in Pacinian afferents can readily induce a vibratory percept with the same low frequency attributes as sinusoidal stimuli of the same frequency, thus demonstrating a universal frequency decoding system. We achieved this using brief low-amplitude pulsatile mechanical stimuli to selectively activate Pacinian afferents. This indicates that spiking pattern, regardless of receptor type, determines vibrotactile frequency perception. This mechanism may underlie the constancy of vibrotactile frequency perception across different skin regions innervated by distinct afferent types.
DOI: https://doi.org/10.7554/eLife.46510.001

*For correspondence:
i.birznieks@unsw.edu.au

Competing interests: The authors declare that no competing interests exist.

## Introduction

The sense of touch comprises a range of different perceptual qualities subserved by several distinct mechanoreceptor types and associated afferent nerve fibres in the skin (*Johnson, 2001*; *Vallbo and Johansson, 1984*). Observations that different receptor types are tuned to different stimulus features and have distinct response profiles have led researchers to conclude that different receptor types are the inputs to separate neural 'channels' dedicated to processing of those features (*Bolanowski et al., 1994*; *Gescheider, 1976*; *Gescheider et al., 2004*; *Hyvarinen et al., 1968*; *Sretavan and Dykes, 1983*). In the glabrous skin, there are two types of fast adapting (FA) afferents which at the threshold level display characteristic U-shaped tuning curves to sinusoidal vibrotactile stimuli: FAI (or RA) afferents innervating Meissner's corpuscles are preferentially activated at frequencies up to 60 Hz, while FAII (or Pacinian - PC) afferents innervating Pacinian corpuscles have much lower response thresholds, and are most sensitive to higher frequencies (>100 Hz) (*Talbot et al., 1968*). At the border between those two frequency domains, at about 60 Hz, there is a qualitative change in sensation from flutter to vibratory hum (*Gescheider, 1976*; *LaMotte and Mountcastle, 1975*; *Talbot et al., 1968*), which is used as further justification for psychophysical segregation into Pacinian and non-Pacinian channels. This scheme has engendered speculation that the Pacinian channel may not possess neural circuits for processing low-frequency spiking patterns

characteristic of low-frequency sinusoidal stimuli, and therefore cannot produce a perceptual experience outside the high-frequency domain. These findings are based on laboratory testing using sinusoidal stimuli in which acceleration and periodicity are linked and thus Pacinian corpuscles would not respond at low frequencies. An intriguing question is the extent to which frequency processing circuitry is specialised for afferent type (FAI vs FAII) in their optimal sinusoidal frequency response range? There is a big gap in our knowledge, as sinusoidal stimuli inherently do not allow activation of FAII afferents at low frequencies and thus functionally are not representative for a wide variety of natural stimuli involving discrete mechanical transients associated with motor control or surface structures with low spatial frequency.

We addressed this question by using brief pulsatile mechanical stimuli that enabled us to create arbitrary time-controlled spike trains of any frequency and pattern in the responding FAII afferents and thus investigate the perceptual properties of those spiking patterns (for details see *Birznieks and Vickery, 2017*). By setting the amplitude of the mechanical pulses below the FAI activation threshold, we first established that low-frequency discharge in FAII afferents (the Pacinian channel) can indeed cause conscious perception of a tactile stimulus at frequencies as low as 6 Hz. We then investigated the perceptual properties of low-frequency FAII afferent discharge by comparing them with those elicited by sinusoidal stimuli driving predominantly FAI afferents (the non-Pacinian or RA channel). We tested whether low-frequency discharge in FAII afferents evoked a clear identifiable percept of frequency and whether it was analogous to that evoked by sinusoidal stimuli within flutter range that primarily activates FAI afferents. Finally, we evaluated frequency discrimination capacity mediated by FAII afferents.

## Results

### Detection thresholds mediated by FAII afferents at low frequencies

Detection thresholds for pulsatile stimuli (*Figure 1b*) evoking low-frequency discharge exclusively in FAII afferents (the Pacinian channel) were measured at two frequencies within the flutter range (6 and 24 Hz) and for comparison at two frequencies in the vibratory hum range (100 and 200 Hz). For pulsatile stimuli, the detection thresholds on the finger were low at all frequencies: 1.3 ($\pm$ 0.6 mean $\pm$ SD; n = 6) µm at the lowest (6 Hz) frequency and 0.7 ($\pm$ 0.2; n = 6) µm at the highest (200 Hz) (*Figure 1a*). Regardless of the frequency, the perceptual thresholds for pulsatile stimuli were well below response threshold for FAI afferents (*Johansson et al., 1982*; *Saal et al., 2017*; *Talbot et al., 1968*), and thus could only have been mediated by the FAII afferents through the Pacinian channel. The detection thresholds for sinusoidal stimuli (*Figure 1b*) were considerably higher within flutter range frequencies and, as expected, steeply decreased with increasing frequency from 28 ($\pm$ 6; n = 6) µm at 6 Hz to 0.7 ($\pm$ 0.2; n = 6) µm at 200 Hz (*Figure 1a*). This reflects a shift from activation of FAI afferents, which have thresholds around 10–15 µm even at their characteristic frequencies, to activation of the much more sensitive FAII afferents which have thresholds for sinusoidal stimulation below 1 µm at their characteristic frequencies (*Johansson et al., 1982*; *Saal et al., 2017*; *Talbot et al., 1968*).

### Perceptual properties of low-frequency discharge rate in Pacinian channel

We next examined whether the Pacinian channel is capable of conveying a sense of vibration frequency within the flutter range. To do this, the amplitude of the pulsatile vibrotactile stimuli was kept at the level of 3 µm regardless of repetition rate (frequency), which is well below the activation thresholds of FAI afferents. The amplitudes for comparison sine waves were selected after conducting intensity matching to the 3 µm pulsatile stimulus in pilot experiments and are presented in *Table 1* of the Materials and methods. Given variability in intensity matching between subjects, we verified that frequency judgements were insensitive to amplitude changes (*Figure 2—figure supplement 1*). The apparent frequency of a 20 or 40 Hz FAII-driven pulsatile stimulus was obtained from participants' comparisons of the pulsatile (P) and sinusoidal (S) stimuli in the following combinations PP, SP, with SS as a control (*Figure 2*). From these comparisons, we calculated the point of subjective equality (PSE) of frequency. The physical frequency defined as repetition rate for pulsatile or frequency for sinusoidal stimuli used as test stimulus was either 20 or 40 Hz.

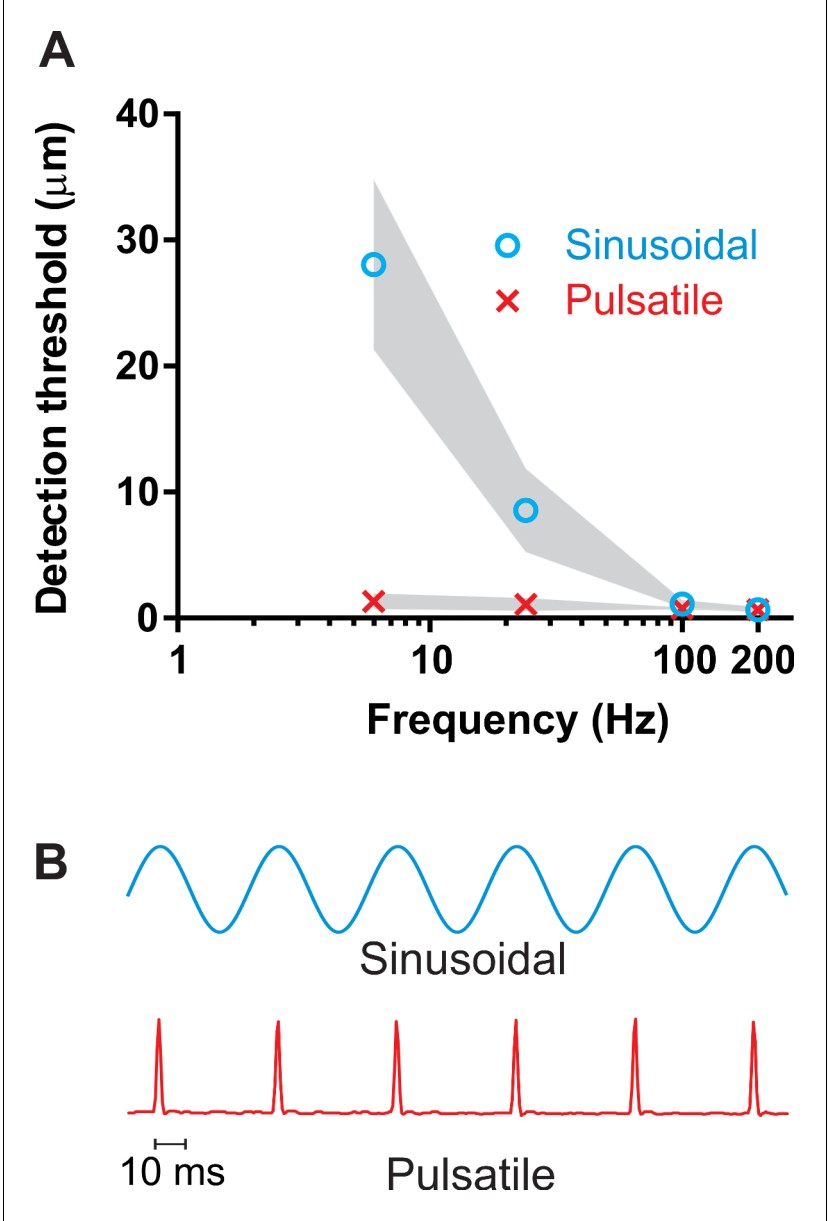

**Figure 1.** Detection thresholds. (**A**) Vibrotactile detection thresholds on the finger across frequency ranges for sinusoidal and pulsatile stimuli (n = 12). Shaded area represent ± 95% confidence intervals. (**B**) An example of the sinusoidal and pulsatile waveforms.

DOI: https://doi.org/10.7554/eLife.46510.002

The following source data is available for figure 1:

**Source data 1.** Detection thresholds.

DOI: https://doi.org/10.7554/eLife.46510.003

### FAII afferent activation in flutter range creates frequency percept

In the PP condition, participants compared the frequency of a pulsatile test stimulus with that of six pulsatile comparison stimuli. The PSEs obtained from the psychometric curves were very close to the physical frequencies of the presented test stimuli: 20.0 (19.5–20.5; 95% confidence interval, CI) Hz for the 20 Hz test, and 40.8 (39.8–41.7) Hz for the 40 Hz test stimulus (*Figure 2*; n = 12). The narrow CI values indicate that pulsatile stimuli evoked perceptions with a well-defined frequency. The PSE values obtained with sinusoidal stimuli (SS condition) using six sinusoidal comparison frequencies

**Table 1.** Amplitudes and frequencies used in each experimental condition.

| Condition | Test stimulus | Comparison stimulus | Comparison frequencies |
|---|---|---|---|
| PP | 20 Hz, Pulsatile, 3 µm<br>40 Hz, Pulsatile, 3 µm | Pulsatile, 3 µm<br>Pulsatile, 3 µm | 10, 14, 18, 22, 26, 30 Hz<br>25, 31, 37, 43, 49, 55 Hz |
| SS | 20 Hz, Sinusoidal, 150 µm<br>40 Hz, Sinusoidal, 40 µm | Sinusoidal, 150 µm<br>Sinusoidal, 40 µm | 10, 14, 18, 22, 26, 30 Hz<br>25, 31, 37, 43, 49, 55 Hz |
| SP | 20 Hz, Sinusoidal, 150 µm<br>40 Hz, Sinusoidal, 40 µm | Pulsatile, 3 µm<br>Pulsatile, 3 µm | 10, 14, 18, 22, 26, 30 Hz<br>25, 31, 37, 43, 49, 55 Hz |

DOI: https://doi.org/10.7554/eLife.46510.007

were 19.5 (18.8–20.2) Hz for the 20 Hz test and 41.7 (39.8–43.7) Hz for the 40 Hz test stimulus (*Figure 2*; n = 12). Repeated measures two-way ANOVA indicated no difference between types of stimuli (pulsatile or sinusoidal) used ($F_{(1, 11)}$=0.131, p=0.72).

## Frequency percept mediated by low-frequency discharge in FAII afferents is analogous to that evoked by sinusoidal stimuli

The PSE for the 20 Hz pulsatile test stimulus was 22.0 (19.7–24.4; 95% CI) Hz when determined in comparison to six sinusoidal frequencies (PS condition), which was no different from the 20 Hz stimulus (p=0.09, n = 12; one sample two-tailed t-test). This indicates that both pulsatile and sinusoidal low-frequency stimuli generate a percept of identical frequency within the flutter frequency range. For 40 Hz pulsatile test stimulus, the PSE assessed in comparison with sinusoidal stimuli was 43.6

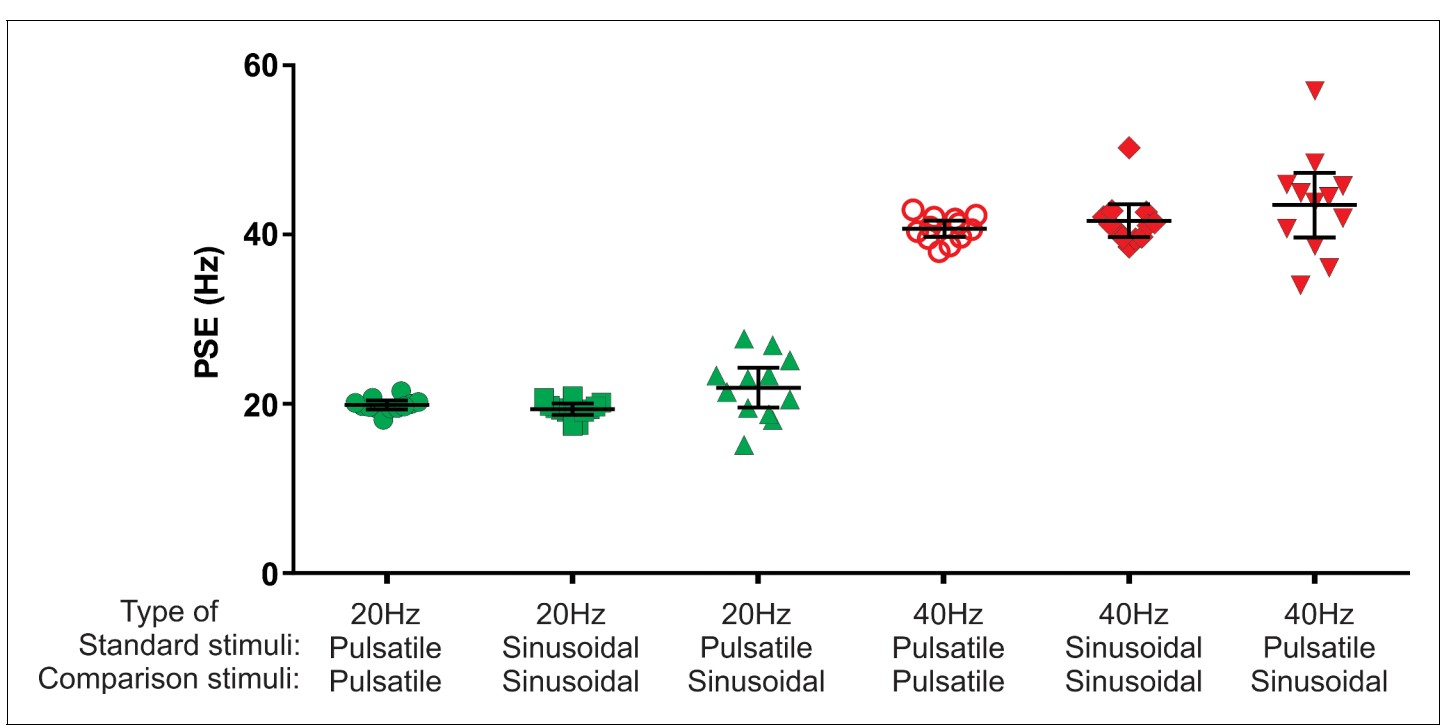

**Figure 2.** Point of subjective equality (PSE) obtained using two interval forced choice paradigm. The test stimulus was either sinusoidal or pulsatile presented at 20 Hz and 40 Hz. The test stimulus was compared with a range of comparison frequencies: 10, 14, 18, 22, 26, 30 Hz with 20 Hz test stimulus; and 25, 31, 37, 43, 49, 55 Hz with 40 Hz test stimulus. Black horizontal lines represent mean ± 95% confidence intervals (n = 12).

DOI: https://doi.org/10.7554/eLife.46510.004

The following source data and figure supplement are available for figure 2:

**Source data 1.** PSE values for individual subjects.

DOI: https://doi.org/10.7554/eLife.46510.006

**Figure supplement 1.** Insensitivity of frequency rating to changes in stimulus amplitude.

DOI: https://doi.org/10.7554/eLife.46510.005

(39.8–47.4; 95% CI) Hz; again, this was not different from 40 Hz (p=0.06, n = 12; one sample two-tailed t-test).

## Frequency discrimination capacity mediated by FAII afferents within the flutter range

Weber fractions that were mediated exclusively by FAII afferents within the flutter frequency range were just as low as the Weber fractions determined with sinusoidal stimuli mediated predominantly by FAI afferents (*Figure 3*). Two-way repeated measures ANOVA indicated that the FAII afferents provided frequency discrimination in the flutter range that was no different from sinusoidal stimuli predominantly mediated by FAI afferents ($F_{(1, 11)}$=0.004, p=0.949). However, there was an effect of frequency ($F_{(1, 11)}$=29.00, p=0.0002) indicating that the size of the Weber fraction is affected by frequency and not afferent type providing this input. Weber fractions were lower at the higher frequency (40 Hz) than they were at 20 Hz for both pulsatile and sinusoidal stimuli (0.21 vs 0.14 and 0.19 vs 0.15, respectively; n = 12).

## Discussion

Our study provides strong evidence that low-frequency discharge of FAII afferents providing input to the Pacinian channel can mediate a clear perception of vibration with easily identifiable and distinguishable frequency characteristics within the flutter range. We also established that frequency

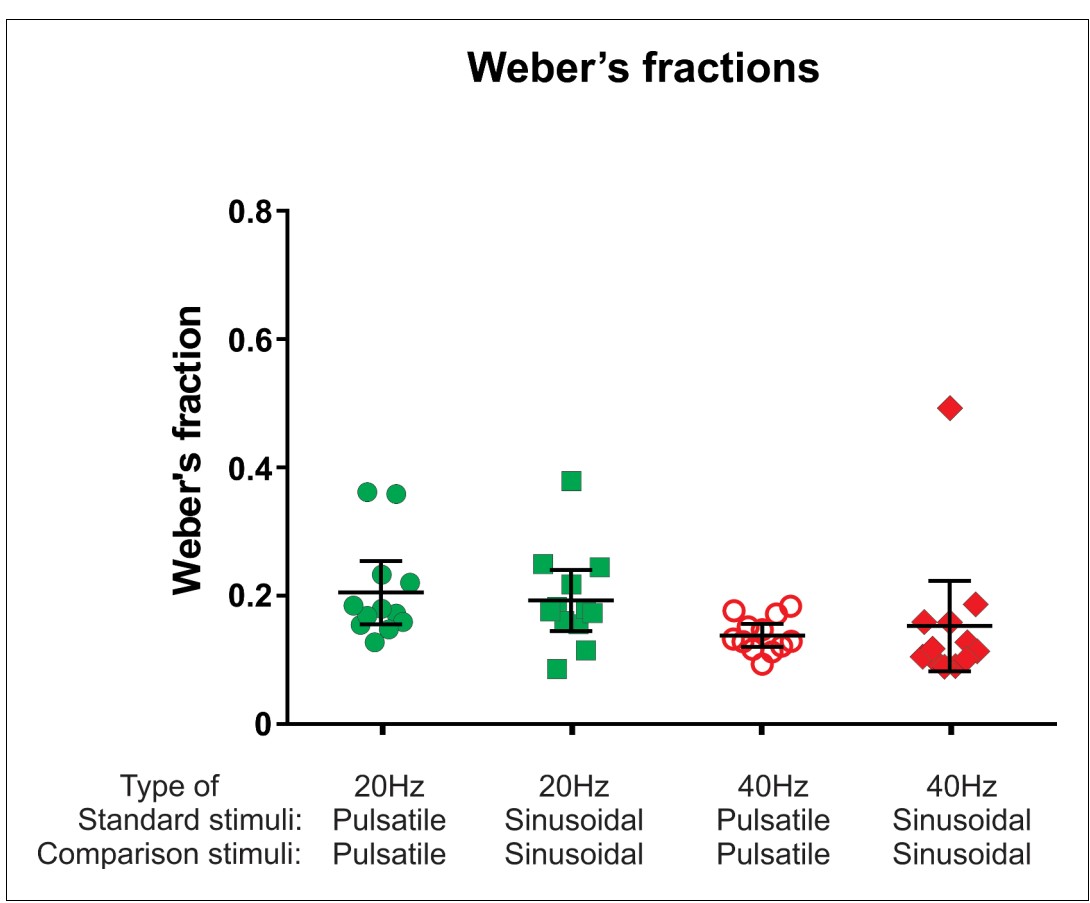

**Figure 3.** The Weber's fraction of just noticeable difference in frequency. For details refer to legend of *Figure 2*.
DOI: https://doi.org/10.7554/eLife.46510.008
The following source data is available for figure 3:

**Source data 1.** Weber's fractions.
DOI: https://doi.org/10.7554/eLife.46510.009

perception signalled exclusively by the activity of FAII afferents is directly comparable with the perceptual features of flutter frequency sensation evoked by corresponding sinusoidal stimuli that naturally activate predominantly FAI afferents (non-Pacinian channel). As the afferent type and thus spatial properties of the population activation evoking this sensation (FAII afferents with large receptive fields) are different from that evoking the same frequency sense with low-amplitude sinusoidal stimulation (FAI afferents with small localised receptive fields), there are likely to be some differences in the quality of sensation beyond the frequency quality. Differences may also relate to the higher synchrony of activation across afferent fibres induced with pulsatile stimuli than with sinusoidal stimuli, due to their tighter time-envelope.

To test whether there are inherent differences in neural mechanisms involved in frequency analysis via the Pacinian and non-Pacinian channels, frequency discrimination ability was tested. The Weber fraction is generally used to characterise the smallest frequency differences reliably detected by subjects as a fraction of the comparison frequency. The Weber fraction is around 0.2 in the flutter range and at high frequencies has been reported to have a slightly higher value of around 0.3; however there is significant variation depending on methodology used (*Bensmaïa et al., 2005*; *Goble and Hollins, 1994*). The discriminative ability of the Pacinian channel was not previously reported at frequencies outside the high-frequency range with which it is normally associated in vibrotactile perception. The Weber fractions determined for the perceived frequency of vibration driven via the FAII and the FAI channels were a close match at both frequencies tested (20 and 40 Hz), and show a similar decrease from 20 to 40 Hz. This suggests that common neural mechanisms of frequency discrimination might be exploited by both channels, as the existence of independent mechanisms having such close agreement at frequencies outside the usual operating range of one of the channels seems less likely.

Recent psychophysical evidence demonstrated that there is interaction between the FAI and FAII inputs, by showing the assimilation effect, where a frequency in the range of one channel can influence perceived frequency on the other channel (*Kuroki et al., 2017*). We suggest that our data extends this, and represents evidence of the functional consequence of the recently discovered extensive convergence of FAI- and FAII- derived inputs onto S1 cortical neurons (*Carter et al., 2014*; *Pei et al., 2009*; *Saal et al., 2015*). It is known that about 70–80% of rapidly adapting neurons in somatosensory cortex S1 show convergent inputs deriving from both afferent classes, and it may be that these neurons are responsible for this generalised frequency processing. *Saal et al. (2015)* made an interesting observation that input from FAI afferents determines the cortical neuron response rate due to its net excitatory drive, while the more temporally-precise PC-channel has a balanced excitatory-inhibitory drive that can control the precise spike timing, which is useful in various encoding schemes (*Birznieks and Vickery, 2017*; *Andrew Hires et al., 2015*; *Johansson and Birznieks, 2004*; *Prsa and Huber, 2018*; *Saal et al., 2016*). This difference did not appear to affect frequency perception in the current study, as Weber fractions were found to be similar regardless of whether we used pulsatile stimuli exclusively activating FAII afferents in a time-controlled manner or used sinusoidal stimuli predominantly activating FAI afferents, presumably with less temporal precision due to the slow rising phase of the sinusoid.

A consequence of generalised neural processing for frequency, regardless of the source of afferent input, is that it would support constancy of vibrotactile frequency perception across different skin regions innervated by different afferent types. For example, the frequency perception on the hairy skin of the arm is not noticeably different from that of the glabrous skin (*Mahns et al., 2006*; *McIntyre et al., 2016*), despite it having neither Meissner (FAI) nor Pacinian (FAII) receptors; instead vibrotactile stimuli are signalled by field units and hair follicle units (*Vallbo et al., 1995*). This also accords with natural stimulation, which is often of a sufficiently high amplitude to activate multiple types of tactile afferents (*Johansson et al., 1982*), and activate receptors across different skin types.

Functionally, it means that FAII afferents and the Pacinian channel are well suited for detecting fast discrete mechanical transients with low repetition rate as might arise during object manipulation or exploration of surfaces with sparsely distributed sharp asperities or ridges. The evidence that low-frequency signals arising from FAII afferents are consciously perceived and easily discriminated strongly suggests that they are biologically important and are likely to be utilised by neural circuits dedicated to motor control of the hand. In regard to new technology development, the exquisite sensitivity of FAII afferents combined with their role in tactile perception and motor control makes them a useful target when designing haptic and teleoperated devices.

## Conclusions

In this study we obtained evidence that low-frequency spike trains in FAII afferents (Pacinian channel) can readily induce a vibratory percept with the same low frequency attributes as signalled by FAI afferents (Meissner's, non-Pacinian channel). It has become evident that perception of vibrotactile frequency depends on the discharge *pattern* of the active afferents, rather than the afferent *type* that is active. Low frequency spike trains in FAII afferents can induce a vibratory percept which has the same frequency attributes as that induced by sinusoidal stimuli. These new findings raise questions about whether much of the observed functional dichotomy between Pacinian and non-Pacinian channels relates to behavioural interpretations of the stimulus rather than to the type of receptor that the signal originates from. In addition, our proposed universal frequency decoding system would help explain the perceptual constancy of vibrotactile frequency perception which is a prominent problem in tactile system where distinct human skin regions and types (e.g. glabrous and hairy) functionally encode the same physical features of stimuli using remarkably different receptor types tuned for different stimulus features.

These findings are consistent with the growing evidence of extensive convergence of inputs from different afferent types onto neurons in the primary somatosensory cortex. Finally, these findings indicate the need to review the functional and neurophysiological basis on which processing of vibrotactile stimuli is attributed to Pacinian and non-Pacinian channels.

# Materials and methods

## Subjects

Research participants were healthy volunteers aged 20 to 26 years without any known history of neurological disorders which would affect the somatosensory system. Ethics approval was obtained from the UNSW Human Research Ethics Committee, and all participants signed a consent form. The participants were reimbursed for their time. There were six participants (four female) in the detection threshold experiments and 12 (6 female) in the frequency perception experiments. Five participants were in both experiments. The sample size was determined by pilot studies to determine effect size, and according to accepted practice in psychophysical experiments. No individual subjects or data outliers were excluded from the data analyses.

## Apparatus

The mechanical stimulation probe was a metal ball 5 mm in diameter at the end of a metal rod driven by a V4 shaker (Data Physics, San Jose). To drive the shaker, analogue output signals were amplified by a Signalforce 30W Power Amplifier (Data physics, San Jose, USA). The displacement of the stimulation probe was monitored using an OptocoNCDT 2200–10 laser displacement sensor (Micro-Epsilon, Ortenburg, Germany) with a resolution of 0.15 μm at 10 kHz.

Stimulus delivery was controlled by a CED data acquisition system (CED, Cambridge, UK) consisting of hardware (CED Power 1401 MkII) and software (Spike2 7.07). Custom made Spike2 and MATLAB (MathWorks Inc, Natick, MA) scripts were used to control the delivery of pulsatile and sinusoidal stimuli, and to record stimulus measurements and button presses made by the participant.

The stimuli were delivered to the finger pad of the right index finger. The arm, hand and stimulated finger were positioned and held in place with the aid of a vacuum pillow (GermaProtec, Kristianstad, Sweden). The pillow, filled with small foam balls, was moulded around the participant's arm, and the air was then pumped out to hold its shape. The probe was positioned on the finger with a force of 50 g; the probe protracted from this rest position. White noise was delivered through headphones to eliminate auditory cues associated with the mechanical stimulator. Participants made responses by pressing buttons with the unstimulated hand.

## Vibration stimulus

A stereotyped brief pulsatile mechanical stimulus with a protraction time of only 2 ms was used to control the spiking pattern in recruited afferents (*Figure 1a*). As the duration of the mechanical stimulus was comparable to the refractory period of the action potential, each mechanical stimulation event generated only a single time-controlled spike in responding afferents (*Birznieks and Vickery,*

2017). Each mechanical pulse is a reproducible and uniform event which ensures that the same population of afferents will be excited regardless of the rate at which these pulses are repeated.

## Detection thresholds

Detection thresholds were measured for pulsatile and sinusoidal stimuli at four frequencies: 6, 24, 100 and 200 Hz. Thresholds were determined on the fingertips. All together thresholds were tested in 16 conditions (2 waveforms x 4 frequencies x 2 locations). The thresholds for two types of stimuli were measured together in a single session with their trials pseudo-randomly interleaved. Each testing session lasted about 10 min, with a total of 8 sessions for each participant.

To measure detection thresholds, we used a two-interval forced-choice (2IFC) procedure, where in each trial participants were presented with two time intervals, indicated with audio cues (*Figure 4a*). The intervals were each 1 s long, with a 0.5 s gap in between. One interval contained the vibration, and the other did not. Participants had to indicate which interval, the first or the second, contained the vibration stimulus. The interval containing the stimulus varied randomly, with each containing the stimulus equally often throughout the experiment.

To calculate detection thresholds, we used the QUEST package implemented in Psychtoolbox-3 (http://psychtoolbox.org) for MATLAB. We defined the threshold as the intensity at which the stimulus could be correctly identified for 82% of trials, and was given by the mean of the posterior distribution function. For each threshold estimate, 41 trials were conducted. To determine the amplitude of the vibration to present on each trial, we used a Bayesian adaptive QUEST protocol (*Watson and Pelli, 1983*), operating on the log-transformed amplitudes. The prior threshold estimate depended on the waveform and frequency (80 μm for 6 Hz sinusoidal, 5 μm for 24 Hz sinusoidal and 3 μm for 100 and 200 Hz sinusoidal, and for all pulsatile stimuli). The amplitude of the vibration on each trial was determined by the QUEST algorithm in most cases. The exceptions were the first trial, which was fixed at the prior threshold estimate for that stimulus, and every tenth trial, which was three

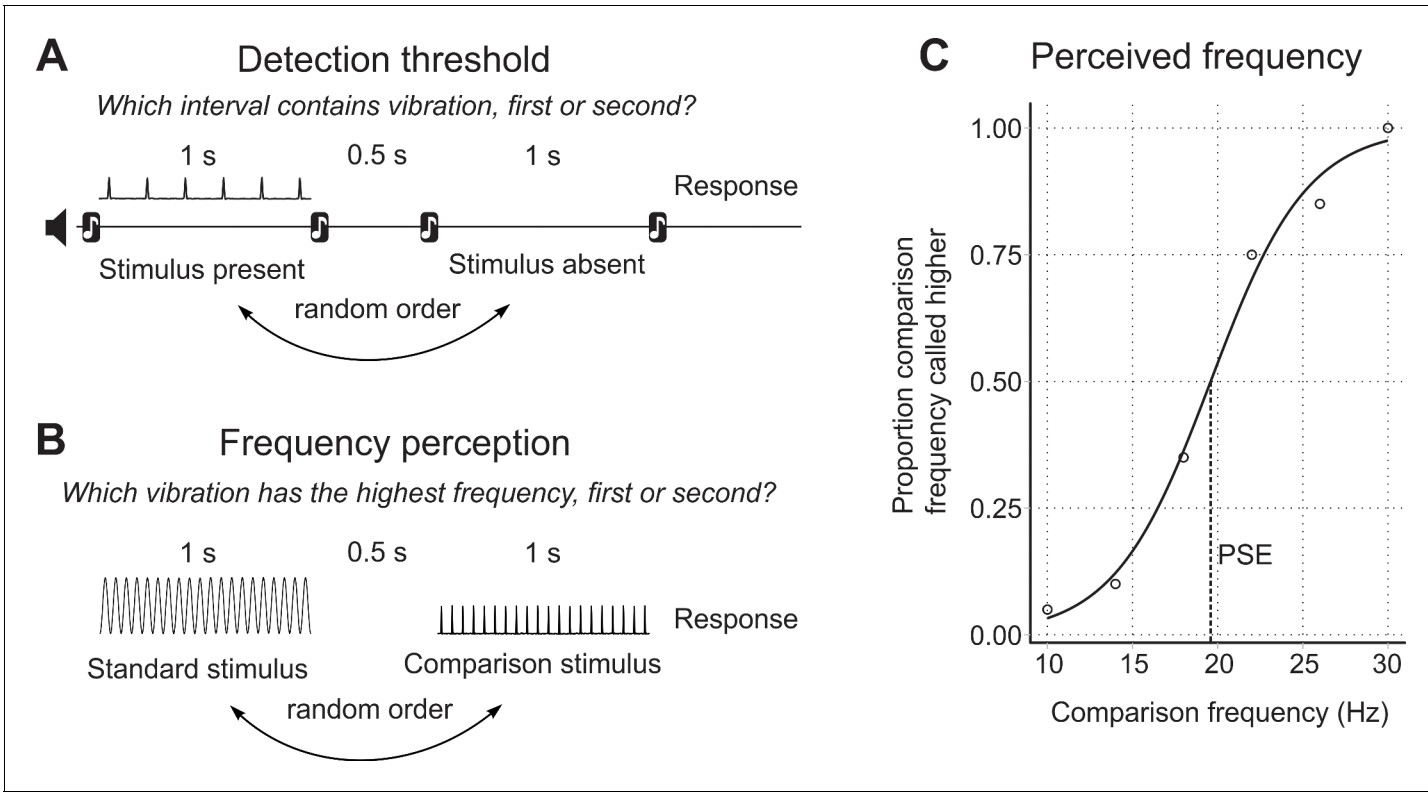

**Figure 4.** Experimental protocols. (**A**) Structure of the detection threshold task. (**B**) Structure of the frequency perception task. (**C**) Point of subjective equality (PSE) determined on the psychometric curve.
DOI: https://doi.org/10.7554/eLife.46510.010

times the value suggested by QUEST, to give participants a few easy trials. The actual amplitude delivered was measured and this value, along with the participant's response, was returned to the QUEST algorithm on each trial.

## Frequency perception

To measure frequency perception, we used a similar 2IFC procedure as described above. However, in this case, both stimulus intervals presented to the participant contained vibration stimulation. The participant was required to indicate which of the two intervals contained the vibration with the highest frequency, the first or the second. One interval contained the test stimulus of fixed frequency, and the other contained the comparison stimulus, which varied in frequency from trial to trial (*Figure 4b*). Six comparison frequencies were paired with the test stimulus 20 times each, in a random sequence.

Frequency perception was tested in three experimental conditions, with different combinations of stimulus waveforms: PP, in which a test stimulus with a pulsatile waveform was compared to comparison stimuli with pulsatile waveforms; SS, in which a sinusoidal test was compared to sinusoidal comparison stimuli; and SP, in which a sinusoidal test was compared to pulsatile comparison stimuli. Each condition was tested with two test stimuli of different frequencies: 20 and 40 Hz. The features of the stimuli used in the experiment are fully described in *Table 1*. For each condition, we randomly interleaved the trials from the 20 and 40 Hz tests. Data collecting from each participant was divided into six sessions lasting approximately 10 min each.

Custom MATLAB scripts were used to analyse the frequency perception data (*McIntyre, 2019*; copy archived at https://github.com/elifesciences-publications/touch-frequency-perception). Logistic regression was applied to the data to produce the psychometric function, relating the frequency of the comparison stimulus to the proportion of trials that the participant said the comparison was a higher frequency than the test. From the psychometric function, we calculated measures of both the perceived frequency of the test stimulus, and the frequency discrimination sensitivity. The perceived frequency of the test stimulus is given by the point of subjective equality (PSE), the comparison frequency at the 50% point on the psychometric curve (*Figure 4c*) (*Birznieks and Vickery, 2017*). The PSE is the point where the participant is equally likely to say the comparison frequency is higher or lower than the test. Discrimination sensitivity is given by the Weber fraction, the one half difference between the 75% point and the 25% on the psychometric function, divided by the frequency of the test stimulus (*LaMotte and Mountcastle, 1975*).

The amplitude of all pulsatile stimuli was 3 μm which was approximately three times the sensory threshold found in our detection threshold experiment. At this amplitude, we expect only Pacinian (FAII) afferents to respond, as threshold for recruitment of FAI afferents, even at their preferred frequency within the flutter range, is no lower than 10 μm. The shape (acceleration profile) of the pulsatile stimulus used in this study corresponds to a waveform of sinusoidal stimulus > 250 Hz. For stimuli at this frequency, FAI afferents typically respond at amplitudes about 10 fold higher than the 3 μm stimuli used for selective FAII afferent activation in this study (*Freeman and Johnson, 1982*; *Johansson et al., 1982*; *Saal et al., 2017*; *Talbot et al., 1968*). The response of one FAII afferent recorded by microneurography in a human subject, evoked by 3 μm pulsatile stimuli at 20 imp/s, is shown in *Figure 5a*. For comparison, the response of one FAI afferent to 20 Hz sinusoidal stimuli also responding at 20 imp/s is shown in *Figure 5b*. The response thresholds with pulsatile stimuli are shown in *Figure 5c–e*. The FAI afferent starts responding sporadically at 30 μm and becomes entrained 1:1 with a stimulus amplitude of 35 μm.

For the sinusoidal stimuli, we chose amplitudes such that the perceived intensity of all stimuli were approximately equal (see *Table 1*). Given that there is inter- and intra- subject variability, we verified that intensity cues were not used for frequency judgements. We conducted control experiments in five subjects (n = 5) who conducted the same experimental protocol illustrated in *Figures 2* and *3* at 40 Hz, but with two different amplitudes (randomly interleaved) for the comparison frequencies. For the sinusoidal comparisons, the standard was 60 μm, and the comparisons were 40 μm (sine low) and 90 μm (sine high). For the pulsatile comparisons, the standard was 6 μm, and the comparisons were 3 μm (pulse low) and 10 μm (pulse high). The data in *Figure 2—figure supplement 1* show that amplitude of the sinusoidal or pulsatile stimuli had no effect on PSE or Weber's fractions indicating that subjects can readily judge frequency while ignoring any intensity cues.

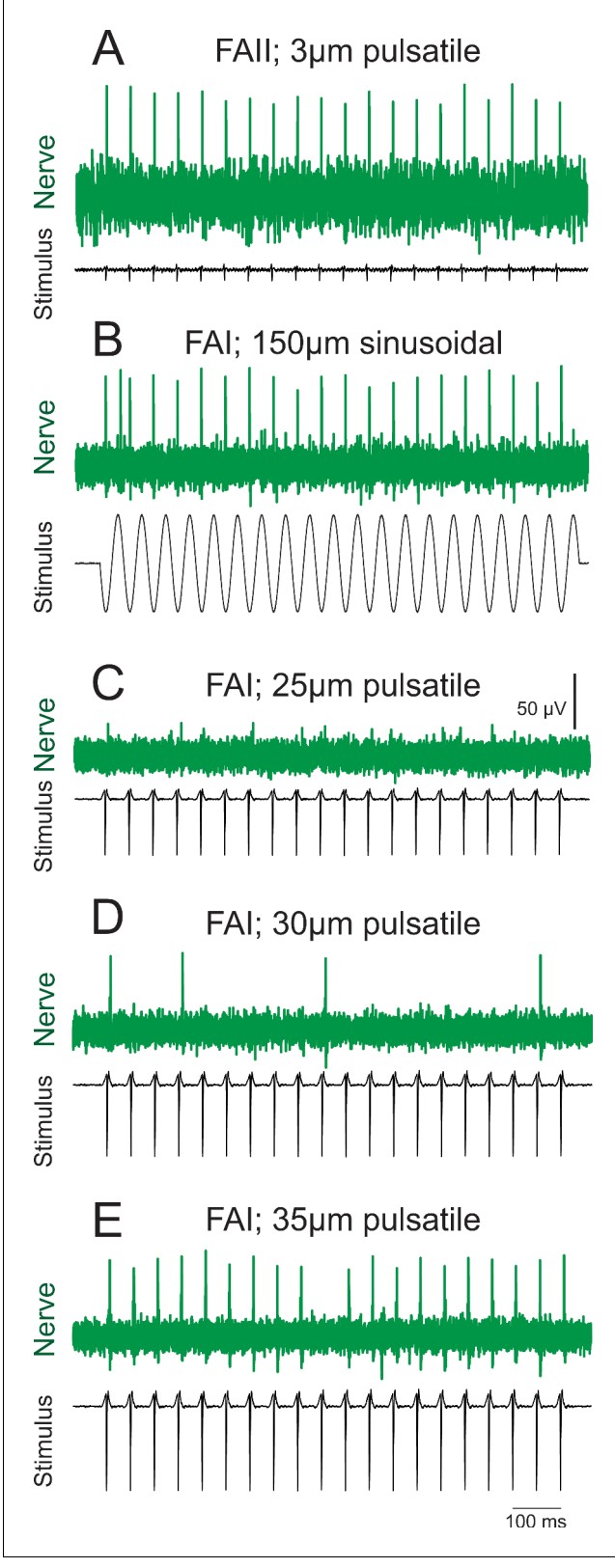

**Figure 5.** Afferent responses with 20 Hz stimuli. (**A**) FAII afferent response to pulsatile stimuli 3 μm in amplitude. (**B**) FAI afferent response to sinusoidal stimuli 150 μm in amplitude. (**C–D**) The same FAI afferent as in B, response to pulsatile stimuli at various amplitudes: no response with 25 μm (subthreshold) stimulus; sporadic firing at 30 μm; and entrainment at 35 μm. Note that firing pattern in A, B and E is identical regardless of stimulus or afferent type.
*Figure 5 continued on next page*

*Figure 5 continued*

DOI: https://doi.org/10.7554/eLife.46510.011

## Statistical analysis

One sample two-tailed t-test (n = 12) was used to test whether the PSE obtained in psychophysics experiments using either pulsatile or sinusoidal stimuli in 12 subjects rendered the same result as physical frequency of the periodic mechanical stimulus of the same type. In this test, PSE obtained by comparing pulsatile stimulus (test stimulus) with sinusoidal stimuli (comparison stimuli) was compared to the expected PSE if test and comparison stimuli would be of the same type (sinusoidal).

Two-way repeated measures ANOVA was performed to analyse the effects on PSE by two repeated measures (within subject; n = 12) factors: type of stimulus (pulsatile, sinusoidal) and frequency (20 Hz, 40 Hz). Two-way repeated measures ANOVA was performed to analyse the effects on Weber's fraction (frequency discrimination capacity) by two repeated measures (within subject; n = 12) factors: type of stimulus (pulsatile, sinusoidal) and frequency (20 Hz, 40 Hz).

For statistical analyses on the calculated thresholds, PSEs and Weber fractions, and for generating graphs, GraphPad Prism software was used (GraphPad Software Inc, La Jolla).

## Acknowledgements

This work was supported by the National Health and Medical Research Council (NHMRC) project grant to IB, RMV, and VGM. We would like to thank Mr Kevin KW Ng for help conducting control experiments investigating the effect of amplitude on frequency perception.

## Additional information

### Funding

| Funder | Grant reference number | Author |
|---|---|---|
| National Health and Medical Research Council | APP1028284 | Ingvars Birznieks Vaughan G Macefield Richard M Vickery |
| Australian Research Council | DP170100064 | Ingvars Birznieks |

The funders had no role in study design, data collection and interpretation, or the decision to submit the work for publication.

### Author contributions

Ingvars Birznieks, Conceptualization, Resources, Software, Formal analysis, Supervision, Funding acquisition, Visualization, Methodology, Writing—original draft, Writing—review and editing; Sarah McIntyre, Conceptualization, Software, Formal analysis, Supervision, Investigation, Visualization, Methodology, Writing—review and editing; Hanna Maria Nilsson, Formal analysis, Investigation, Visualization, Writing—original draft, Writing—review and editing; Saad S Nagi, Conceptualization, Formal analysis, Investigation, Writing—review and editing; Vaughan G Macefield, Formal analysis, Funding acquisition, Writing—review and editing; David A Mahns, Formal analysis, Supervision, Methodology, Writing—review and editing; Richard M Vickery, Conceptualization, Resources, Software, Formal analysis, Supervision, Funding acquisition, Visualization, Methodology, Writing—review and editing

### Author ORCIDs

Ingvars Birznieks (iD) https://orcid.org/0000-0003-4916-1254
Saad S Nagi (iD) http://orcid.org/0000-0001-8773-8232

## Ethics

Human subjects: Ethics approval was obtained from the UNSW Human Research Ethics Committee HC11074, HC16245 and all participants signed a consent form.

## Decision letter and Author response

Decision letter https://doi.org/10.7554/eLife.46510.014
Author response https://doi.org/10.7554/eLife.46510.015

## Additional files

### Supplementary files

• Transparent reporting form
DOI: https://doi.org/10.7554/eLife.46510.012

### Data availability

Data analysed during this study are included in the manuscript and supporting files. Source data files have been provided for Figures 1, 2 and 3.

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
