## [Decision Letter]

Thank you for submitting your article "Tactile sensory channels over-ruled by frequency decoding system that utilizes spike pattern regardless of receptor type" for consideration by *eLife*. Your article has been reviewed by three peer reviewers, one of whom is a member of our Board of Reviewing Editors, and the evaluation has been overseen by Andrew King as the Senior Editor. The following individual involved in review of your submission has agreed to reveal their identity: Stefan G Lechner (Reviewer #3).

The reviewers have discussed the reviews with one another and the Reviewing Editor has drafted this decision to help you prepare a revised submission.

Summary:

Birznieks et al. report a simple but clever series of psychophysical experiments to test whether the Pacinian corpuscle-associated (PC) "channel" of tactile afferents is capable of producing perceptions of low-frequency periodic stimuli. Pacinian corpuscles impose a mechanical high-pass filter between skin deflections and mechanotransduction. Accordingly, classic work has linked PC channel activity to the perception of high-frequency periodic stimuli. The present study provides evidence that PC activation can elicit perceptions of low-frequency stimulus periodicity.

Essential revisions:

1) A major issue for all three reviewers concerns the extent to which the low-amplitude pulsatile mechanical stimuli used selectively activate PC afferents and not other afferent types. The authors state that the perceptual thresholds for pulsatile stimuli were well below response threshold for Meissner afferents. Nerve recordings generated using both pulsatile and sinusoidal mechanical stimuli at different frequencies should be done using techniques described in one of the lab's previous papers (Birznieks and Vickery, 2017) and reported in this study.

2) An important question is whether the perceived intensity of a 3 micrometer 20 Hz pulsatile stimulus is the same as the perceived intensity of a 150 micrometer 20 Hz sinusoidal stimulus. As the authors state in the Discussion "…80% of the rapidly-adapting neurons in somatosensory cortex S1 show convergent inputs deriving from both afferent classes….". If a 3 µm 20 Hz pulsatile stimulus and a 150 µm sinusoidal stimulus evoke the same afferent message in different afferents, which, however, project to the same cortical neurons, how is intensity then encoded? It should be straightforward to ask subjects to report differences in perceived intensity. The authors should perform this experiment.

3) The authors should point out to readers in the Discussion that while the pulsatile stimuli contain low-frequency periodicity, each transient is itself high-frequency and may feel quite distinct from anything that could be elicited with the activation of Meissner afferents. Indeed because each pulsatile stimulus is a single cycle of a high-frequency stimulus, the authors do not provide evidence one way or the other about perceptions induced by PC afferent activation in the absence of high-frequency mechanical content (which presumably can only be achieved by bypassing the end organ with microstimulation). This issue needs to be addressed.

4) In the Introduction, the authors state "[o]ne assumption of this scheme is that the Pacinian channel does not possess neural circuits for processing low-frequency spiking patterns characteristic of low frequency sinusoidal stimuli and therefore cannot produce a perceptual experience outside the high frequency domain." The reviewers are not convinced that this is an assumption often made. The more common assumption may be that the PC end organs only allow transduction of high-frequency stimuli, limiting the normal engagement (but not necessarily capabilities) of the downstream PC-linked circuits to processing of high-frequency stimuli. Please discuss this point.

5) It is well known that both Meissner afferents and Pacinian afferent are velocity detectors – i.e. they only fire action potentials during the dynamic phase of a mechanical stimulus. Moreover, it is well known that both afferent subtypes exhibit an ON and an OFF response, that is they fire action potentials both during the forward and the backward movements of a mechanical stimulus. Accordingly, a 20 Hz sinusoidal vibrotactile stimulus triggers a spike train with a frequency of app. 40 Hz (a very nice example for this can be found in Figure 1 in Muniak et al., 2007, J. Neurosci.). By contrast the pulsatile stimuli used here have a protraction time of only 2 ms and therefore – as the authors state – "generated only a single time-controlled spike in responding afferents". Accordingly, the 20 Hz pulsatile stimulus used by the authors evokes a 20 Hz spike train, whereas the 20 Hz sinusoidal stimulus most likely produced a 40 Hz spike train. Thus, a surprising observation was that the 20 Hz sinusoidal and the 20 Hz pulsatile stimuli seem to produce the same sensory percept (Figure 2 pulsatile vs. sinusoidal), despite the fact that they most likely produce different action potential firing patterns. The authors should discuss this point.

6) The description of the method is difficult to understand for non-expert readers. It would be helpful if the authors showed at least one example of a psychometric curve used to determine the PSE.

---

## [Author Response]

Essential revisions:1) A major issue for all three reviewers concerns the extent to which the low-amplitude pulsatile mechanical stimuli used selectively activate PC afferents and not other afferent types. The authors state that the perceptual thresholds for pulsatile stimuli were well below response threshold for Meissner afferents. Nerve recordings generated using both pulsatile and sinusoidal mechanical stimuli at different frequencies should be done using techniques described in one of the lab's previous papers (Birznieks and Vickery, 2017) and reported in this study.

FAI afferents also referred to as RAs best respond to stimuli which have similar dynamic properties of skin indentation as sinusoidal stimuli at frequencies about 30-40 Hz. Within this range of highest FAI afferent sensitivity, the corresponding human detection thresholds are above 10 µm, indicating that FAI cannot evoke any sensation below 10 µm amplitude. At higher sinusoidal frequencies, the thresholds of FA1 afferents increase steeply (see Talbot et al., 1968, Figure 21), thus a considerably larger amplitude stimuli would be required for FAI afferents to contribute. The shape (acceleration profile) of the 3-4 ms duration pulsatile stimulus used in our study corresponds to a waveform of a sinusoidal stimulus of frequency > 250 Hz. For stimuli at this frequency, FAI afferent thresholds are about 3 fold higher than at their best frequency. In a study systematically assessing vibrotactile afferent firing at different frequencies using microneurography recordings in humans by Johansson et al. (1982), none of the recorded FAI afferents responded to 256 Hz vibration at 32 μm amplitude.

In our own microneurography experiments with various pulsatile stimuli we haven't encountered FAI units that respond below these limits (and we now include evidence in the manuscript to support this). Thus there is absolutely no evidence in the experimental literature, or in modelling studies (e.g.TouchSim, Saal et al., 2017), or in our own experience, which would indicate that any FAI afferent would be capable of responding to the 3 µm stimuli used in our study.

We have now added a new Figure 5, and accompanying text, explaining this: “The amplitude of all pulsatile stimuli was 3 µm which was approximately 3 times the sensory threshold found in our detection threshold experiment. […] The response thresholds with pulsatile stimuli are shown in Figure 5C-E. The FAI afferent starts responding sporadically at 30 μm and becomes entrained 1:1 with a stimulus amplitude of 35 μm.”

2) An important question is whether the perceived intensity of a 3 micrometer 20 Hz pulsatile stimulus is the same as the perceived intensity of a 150 micrometer 20 Hz sinusoidal stimulus. As the authors state in the discussion "…80% of the rapidly-adapting neurons in somatosensory cortex S1 show convergent inputs deriving from both afferent classes….". If a 3 µm 20 Hz pulsatile stimulus and a 150 µm sinusoidal stimulus evoke the same afferent message in different afferents, which, however, project to the same cortical neurons, how is intensity then encoded? It should be straightforward to ask subjects to report differences in perceived intensity. The authors should perform this experiment.

The amplitudes for the comparison sine waves presented in Table 1 of the Materials and methods were selected after amplitude matching in pilot experiments to the 3 µm pulsatile stimulus. Given that there is inter- and intra- subject variability in the best match, we have now verified that frequency judgements were insensitive to amplitude changes, and present this data in the Results and in new Figure 2—figure supplement 1. In our experiments we analysed the point of subjective equality for frequency and the Weber fraction at 40 Hz, when two different amplitudes (randomly interleaved) for the comparison frequencies were compared with the standard stimulus.

We now have amended the following paragraph of the Results section:

“The amplitudes for comparison sine waves were selected after conducting intensity matching to the 3 µm pulsatile stimulus in pilot experiments and are presented in Table 1 of the Materials and methods. […] The apparent frequency of a 20 or 40 Hz FAII-driven pulsatile stimulus was obtained from participants’ comparisons of the pulsatile (P) and sinusoidal (S) stimuli in the following combinations PP, SP, with SS as a control (Figure 2).”

We now have added the following paragraph to the Materials and methods section:

“Given that there is inter- and intra- subject variability, we verified that intensity cues were not used for frequency judgements. […] The data in Figure 2—figure supplement 1 show that amplitude of the sinusoidal or pulsatile stimuli had no effect on PSE or Weber’s fractions indicating that subjects can readily judge frequency while ignoring any intensity cues.”

3) The authors should point out to readers in the Discussion that while the pulsatile stimuli contain low-frequency periodicity, each transient is itself high-frequency and may feel quite distinct from anything that could be elicited with the activation of Meissner afferents. Indeed because each pulsatile stimulus is a single cycle of a high-frequency stimulus, the authors do not provide evidence one way or the other about perceptions induced by PC afferent activation in the absence of high-frequency mechanical content (which presumably can only be achieved by bypassing the end organ with microstimulation). This issue needs to be addressed.

We have added the following sentences to the first paragraph of the Discussion:

“Our study provides strong evidence that low frequency discharge of FAII afferents providing input to the Pacinian channel can mediate a clear perception of vibration with easily identifiable and distinguishable frequency characteristics within the flutter range. […] Differences may also relate to the higher synchrony of activation across afferent fibres induced with pulsatile stimuli than with sinusoidal stimuli, due to their tighter time-envelope.”

4) In the Introduction, the authors state "[o]ne assumption of this scheme is that the Pacinian channel does not possess neural circuits for processing low-frequency spiking patterns characteristic of low frequency sinusoidal stimuli and therefore cannot produce a perceptual experience outside the high frequency domain." The reviewers are not convinced that this is an assumption often made. The more common assumption may be that the PC end organs only allow transduction of high-frequency stimuli, limiting the normal engagement (but not necessarily capabilities) of the downstream PC-linked circuits to processing of high-frequency stimuli. Please discuss this point.

If there would be consensus that the same pathways and neural circuits would support physiological effects attributed to two afferent types then there would be little reason to refer to two distinct channels as there would be a continuum of effects at different frequencies. However, we absolutely agree that with new evidence there is a shift in thinking which we discuss in the third paragraph of the Discussion section.

We have now rephrased the statement in Introduction, to say:

“This scheme has engendered speculation that the Pacinian channel may not possess neural circuits for processing low frequency spiking patterns characteristic of low frequency sinusoidal stimuli, and therefore cannot produce a perceptual experience outside the high frequency domain.”

We further explain that this has been limitation of using sinusoidal stimuli in the laboratory settings not capable of activating FAII afferents at low frequencies:

“… and thus functionally are not representative for a wide variety of natural stimuli involving discrete mechanical transients associated with motor control or surface structures with low spatial frequency.”

5) It is well known that both Meissner afferents and Pacinian afferent are velocity detectors – i.e. they only fire action potentials during the dynamic phase of a mechanical stimulus. Moreover, it is well known that both afferent subtypes exhibit an ON and an OFF response, that is they fire action potentials both during the forward and the backward movements of a mechanical stimulus. Accordingly, a 20 Hz sinusoidal vibrotactile stimulus triggers a spike train with a frequency of app. 40 Hz (a very nice example for this can be found in Figure 1 in Muniak et al., 2007, J. Neurosci.). By contrast the pulsatile stimuli used here have a protraction time of only 2 ms and therefore – as the authors state – "generated only a single time-controlled spike in responding afferents". Accordingly, the 20 Hz pulsatile stimulus used by the authors evokes a 20 Hz spike train, whereas the 20 Hz sinusoidal stimulus most likely produced a 40 Hz spike train. Thus, a surprising observation was that the 20 Hz sinusoidal and the 20 Hz pulsatile stimuli seem to produce the same sensory percept (Figure 2 pulsatile vs. sinusoidal), despite the fact that they most likely produce different action potential firing patterns. The authors should discuss this point.

Thank you, this is very interesting question and valid observation. Indeed this is what motivated and inspired the type of studies we have been conducting with pulsatile stimuli. As seen in the new Figure 5B, the FAI afferent responds with 1 spike per sinusoidal cycle at 20 Hz 150 µm. At the beginning of entrainment the firing is 1:1 over an extended range of amplitudes (see Figure 1 in Freeman and Johnson, 1982). Only when amplitude is made sufficiently large, will a second spike be generated, which could be during protraction or retraction phase, and eventually as amplitude increases even more spikes may be generated (see Figure 3 from the follow up paper on intensity encoding from Bensmaia’s lab [Bensmaia, 2008, Behav. Brain Res.]). But these firing pattern changes do not cause frequency perception to double, triple or quadruple at these larger amplitudes. How can the number of spikes encode frequency then? This is exactly what has puzzled researchers in the field and this is a question we were able to answer in our previous Birznieks and Vickery (2017) paper using pulsatile stimuli. It is the temporal pattern of spiking activity that matters and not the spike count. The findings in this current manuscript are further evidence which extends to various afferent types, that spiking pattern determines perceived frequency and not the afferent type activated.

6) The description of the method is difficult to understand for non-expert readers. It would be helpful if the authors showed at least one example of a psychometric curve used to determine the PSE.

We now have included one of the psychometric curves, with explanation, as an example in Figure 4C.